# Asexual reproduction reduces transposable element load in experimental yeast populations

Jens Bast[1†]*, Kamil S Jaron[1,2†], Donovan Schuseil[1], Denis Roze[3,4], Tanja Schwander[1]

[1]Department of Ecology and Evolution, University of Lausanne, Lausanne, Switzerland; [2]Swiss Institute of Bioinformatics, Lausanne, Switzerland; [3]UMI3614 Evolutionary Biology and Ecology of Alga, Centre National de la Recherche Scientifique (CNRS), Roscoff, France; [4]Station Biologique de Roscoff, Sorbonne Université, Roscoff, France

**Abstract** Theory predicts that sexual reproduction can either facilitate or restrain transposable element (TE) accumulation by providing TEs with a means of spreading to all individuals in a population, versus facilitating TE load reduction via purifying selection. By quantifying genomic TE loads over time in experimental sexual and asexual *Saccharomyces cerevisiae* populations, we provide direct evidence that TE loads decrease rapidly under asexual reproduction. We show, using simulations, that this reduction may occur via evolution of TE activity, most likely via increased excision rates. Thus, sex is a major driver of genomic TE loads and at the root of the success of TEs.

DOI: https://doi.org/10.7554/eLife.48548.001

*For correspondence:
mail@jensbast.com

†These authors contributed equally to this work

Competing interests: The authors declare that no competing interests exist.

## Introduction

Self-replicating transposable elements (TEs) can occupy large fractions of genomes in organisms throughout the tree of life (reviewed in *Hua-Van et al., 2011*). Their overwhelming success is driven by their ability to proliferate independently of the host cell cycle via different self-copying mechanisms involving 'cut-and-paste' or 'copy-and-paste' systems. These mechanisms allow TEs to invade genomes in a similar way to parasites, despite generally not providing any advantage to the individual carrying them (*Doolittle and Sapienza, 1980*; *Orgel and Crick, 1980*). To the contrary, TEs generate deleterious effects in their hosts by promoting ectopic recombination and because most new TE insertions in coding or regulatory sequences disrupt gene functions (*Finnegan, 1992*; *Montgomery et al., 1991*).

Theory predicts that sexual reproduction can either facilitate or restrain the genomic accumulation of TEs, and it is currently unclear whether the expected net effect of sex on TE loads is positive or negative. Sexual reproduction can facilitate the accumulation of TEs because it allows TEs to colonise new genomes and spread throughout populations (*Hickey, 1982*; *Zeyl et al., 1996*). Because the colonisation of new genomes is more likely for active TEs, sexual reproduction should favour the evolution of highly active TEs (*Charlesworth and Langley, 1986*; *Hickey, 1982*), even though increased activity generates higher TE loads in the host genome. At the same time, sexual reproduction facilitates the evolution of host defences and increases the efficacy of purifying selection against deleterious TE copies by reducing selective interference among loci (*Ågren and Wright, 2011*; *Arkhipova and Meselson, 2005*; *Crespi and Schwander, 2012*; *Wright and Finnegan, 2001*). In the absence of sex, reduced purifying selection can thus result in the accumulation of TEs, unless TE

**eLife digest** The genetic information of most living organisms contains parasitic invaders known as transposable elements. These genetic sequences multiply by copying and pasting themselves through the genome, but this process can disrupt the activity of important genes and put the organism at risk.

How transposable elements proliferate in a population depends on the way organisms reproduce. If they simply clone themselves asexually, the selfish elements cannot spread between the different clones. If the organisms mate together their respective transposable elements get mixed, which helps the sequences to spread more easily and to potentially become more virulent. However, sexual reproduction also comes with mechanisms that keep transposable elements in check.

Bast, Jaron et al. took advantage of the fact that yeasts can reproduce with or without mating to explore whether sexual or asexual organisms are better at controlling the spread of transposable elements. The number of copies of transposable elements in the genomes of yeast grown sexually or asexually was assessed. The results showed that sexual populations kept constant numbers of selfish elements, while asexual organisms lost these genomic parasites over time. Simulations then revealed that this difference emerged because a defense gene that helps to delete transposable elements was spreading more quickly in the asexual group.

The work by Bast, Jaron et al. therefore suggests that sex is responsible for the evolutionary success of transposable elements, while asexual populations can discard these sequences over time. Sex therefore helps genetic parasites, somewhat similar to sexually transmitted diseases, to spread between individuals and remain virulent.

DOI: https://doi.org/10.7554/eLife.48548.002

copies get eliminated via excision at sufficiently high rates (*Burt and Trivers, 2006*; *Dolgin and Charlesworth, 2006*).

Genomic TE loads have been empirically estimated for natural populations of asexual and related sexual organisms, but no consistent difference emerges (*Ågren et al., 2015*; *Bast et al., 2016*; *Jiang et al., 2017*; *Szitenberg et al., 2016*), probably because many confounding factors not related to reproductive mode such as hybridisation and polyploidisation can affect TE loads and mask the effect of sex (*Arkhipova and Rodriguez, 2013*; *Hua-Van et al., 2011*).

Here, to quantify whether the net effect of sexual reproduction on genomic TE loads is positive or negative, we study the evolution of genomic TE loads in experimental yeast (*Saccharomyces cerevisiae*) populations generated in a previous study (*McDonald et al., 2016*). McDonald et al. maintained four sexual and four asexual strains originating from the same haploid ancestral strain (W303) under constant conditions over 1000 generations. For sexual strains, a mating event (meiosis) was induced every 90 generations. Sequencing of each strain was conducted at generation 0 and every 90 generations prior to mating (for details see Materials and methods, and *McDonald et al., 2016*). In the present study, we use the published Illumina data to quantify TE loads in each strain for each sequenced generation.

TEs in *S. cerevisiae* are well characterised (*Carr et al., 2012*; *Castanera et al., 2016*; *Voytas and Boeke, 1992*). *S. cerevisiae* TEs consist solely of 'copy-and-paste' elements that are flanked by long terminal repeats (LTRs) and are grouped into the families *Ty1-Ty5* (*Voytas and Boeke, 1992*). The 12.2 Mb genome of the studied yeast strain comprises approximately 50 full-length, active *Ty* element copies, and 430 inactive ones (*Carr et al., 2012*). Inactive copies include truncated elements as well as remnants from TE excisions (i.e., solo-LTR formation; *Carr et al., 2012*). Excisions occur by intra-chromosomal recombination between the two flanking LTRs of a TE, and result in the removal of protein-coding genes that allow for transposition.

Using different computational approaches to quantify genomic TE loads in experimental yeast strains, we show that sex is required for the success of TEs, as TE loads decrease over time under asexual reproduction. For the first approach, we quantified total TE loads without distinguishing between active and inactive TEs. This was done by computing the fraction of reads that mapped to a curated *S. cerevisiae* TE library (see Materials and methods) for each yeast strain and sequenced

generation. This analysis revealed that the total TE load in sexual strains remained constant over 1000 generations, but decreased in asexual strains over time (resulting in a total reduction of 23.5% after 1000 generations; generation effect p<0.001, reproductive mode effect p=0.081, and interaction between generation and mode p<0.001; permutation ANOVA, *Figure 1—figure supplement 1*). For the second approach, we focused on full-length TE copy insertions, because only those are active and can lead to increased genomic TE loads over time. Detecting specific TE insertions by aligning short-read data to a reference genome is difficult and associated with a detection bias towards TEs present in the reference genome. Moreover, because sequencing was done with population pools and not individual clones within populations, it is not possible to analyse turnover or activity of TEs within specific genomic backgrounds. Instead, we analysed the presence versus absence of specific TE insertions in each population over time. With a pipeline that combines different complementary approaches (*Nelson et al., 2017*, see Materials and methods), the available sequencing data allowed us to detect 24 out of the 50 full-length insertions that are present in the reference genome of the ancestral strain at the start of the experiment (generation 0). As with the first approach, we found that the number of (detectable) full-length TE copies remained constant in sexual yeast strains, but decreased in asexual strains over time (generation effect p=0.006, reproductive mode effect p=0.033, and interaction between generation and mode p<0.001; permutation ANOVA). In asexual strains, the estimated average number of full-length TEs decreased from approximately 50 to 41 over 1000 generations (*Figure 1*).

This decrease could be generated by either increased TE excision rates in asexual as compared to sexual yeast, reduced transposition rates, or a combination of both mechanisms. To evaluate the relative importance of the two mechanisms, we estimated the number of losses of TEs present in the ancestral yeast strain, as well as the number of novel insertions, at each assayed generation (*Figure 2*). These analyses revealed that 'ancestral' TE insertions are lost at a higher rate in asexual than sexual strains (generation effect p=0.002, reproductive mode effect p=0.027, and interaction between generation and mode p<0.001; permutation ANOVA), while we detected similar numbers of novel TE insertions (indicating similar transposition rates) under both reproductive modes (generation effect p=0.338, reproductive mode effect p=0.271, and interaction between generation and mode p=0.599; permutation ANOVA). Taken together, our empirical observations indicate that even very rare events of sex (here just 10 out of 990 reproduction events) are sufficient to maintain genomic TE loads, while asexuality results in the reduction of TE loads.

The parallel reduction of TE loads in different asexual strains suggests that the evolution of reduced TE activity (the ratio of transposition to excision) in asexual strains influences genomic TE loads more strongly than purifying selection, which should act to reduce TE loads most effectively in sexual strains. To evaluate whether these findings are plausible, we tested whether the net loss of TEs under asexualilty is predicted by a simple model of TE dynamics. As explained above, different theoretical approaches have shown that both purifying selection and activity rate evolution can affect TE loads under sexual or asexual reproduction (*Charlesworth and Langley, 1986*; *Dolgin and Charlesworth, 2006*; *Hickey, 1982*). However, no theoretical study has considered TE load evolution under the joint effects of the different processes. To fill this gap, we extended the individual-level simulation program of *Dolgin and Charlesworth (2006)*. This program allows to study the evolution of TE copy numbers in an asexual lineage as a function of TE activity (the joint effects of transposition and excision rates), as well as of the strength of selection against TE insertions, which depends on the fitness cost per TE insertion. To compare TE loads in sexual and asexual lineages, we first extended the program to include events of sexual reproduction and parameterised the simulations with empirically determined values from yeast (*Blanc and Adams, 2004*; *Carr et al., 2012*; *Garfinkel et al., 2005*). We ran individual-based simulations with a range of transposition rates, excision rates and selection coefficients with and without epistasis between TE copies as pertinent for yeast (see *Supplementary file 2A*).

For all simulations, TE loads in populations undergoing sex every 90 generations decreased faster than in asexual populations, contrary to our empirical observations. This occurs because sexual events generate variation among individuals in TE loads (and thus variation in fitness), which facilitates selection against deleterious TEs (see also *Dolgin and Charlesworth, 2006*). Different transposition rates under meiosis (sex) or mitosis (asex) did not affect this finding. Indeed, increased TE activity during meiosis only transiently increases TE loads in sexual strains. Because such activity also generates increased variation in TE loads (and therefore in fitness) among strains, the additional TE

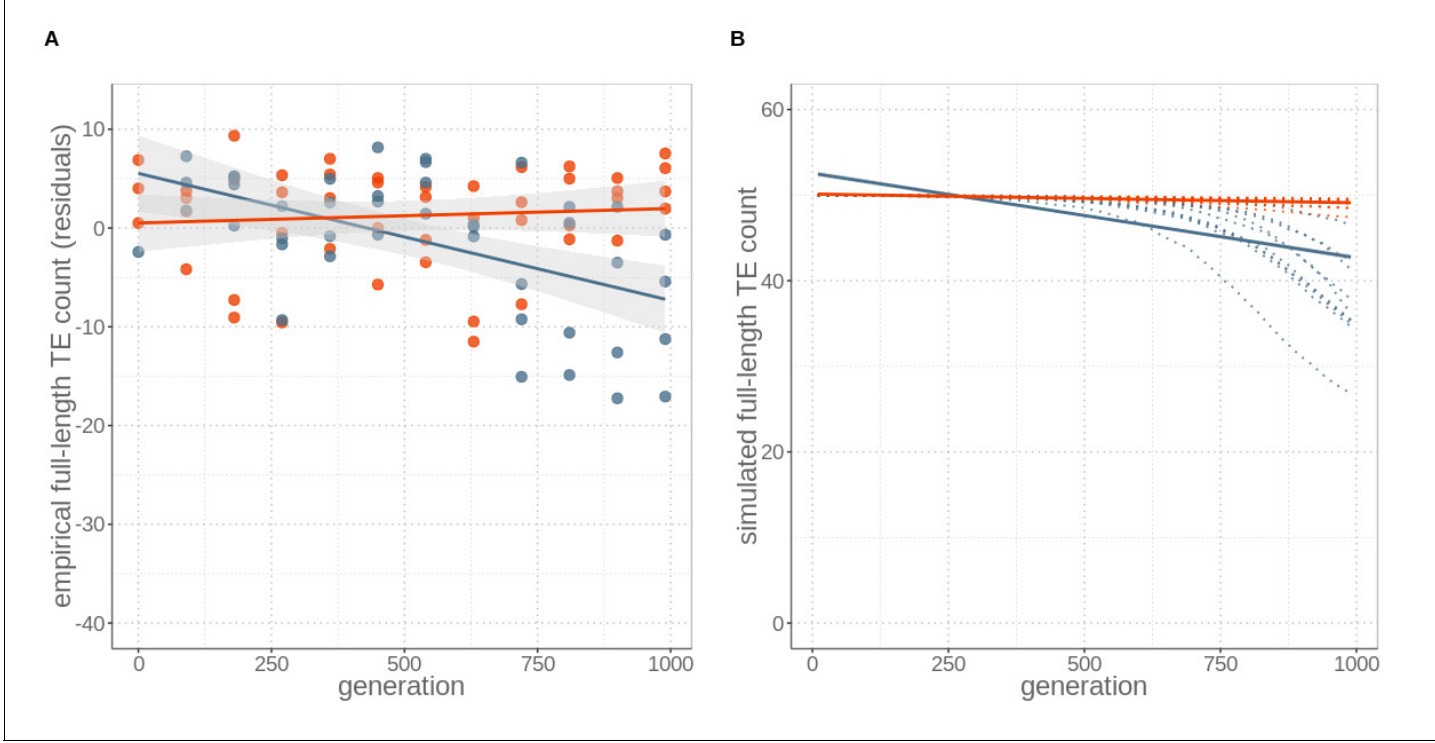

**Figure 1.** Sex maintains constant TE loads through time, while its absence leads to TE copy number reductions, for both (A) empirical data and (B) simulations including an allele modifying TE activity rates. (A) Number of full-length TE copies inserted in genomes of four replicates of otherwise identical occasionally sexual (red) and wholly asexual (blue) yeast strains over 1000 generations of experimental evolution. Numbers are expressed as residuals, since the TE detection probability depends on sequencing coverage (*Figure 1—figure supplement 2*). (B) Individual-based simulations for studying the TE load dynamics expected under sexual and asexual reproduction with ten replicates (red and blue dotted lines). The simulations are parameterised with yeast-specific values and include a modifier alleles. For both (A) empirical and (B) simulation data, asexuals lost about nine active, full-length TEs by generation 1000. Lines represent linear regression for sexuals (red) and asexuals (blue) and the grey areas represent 95% CI.

DOI: https://doi.org/10.7554/eLife.48548.003

The following figure supplements are available for figure 1:

**Figure supplement 1.** Overall transposable element load remains stable in sexual strains, but is reduced in asexual strains after 1000 generations.

DOI: https://doi.org/10.7554/eLife.48548.004

**Figure supplement 2.** Identification of TE insertions depends on the sequencing coverage.

DOI: https://doi.org/10.7554/eLife.48548.005

**Figure supplement 3.** Simulations with higher transposition rates during meiosis than mitosis.

DOI: https://doi.org/10.7554/eLife.48548.006

**Figure supplement 4.** In the simulations, the spread of a modifier of excision rates is faster in asexual than sexual populations because it remains linked to genomes that have few TE copies and therefore a high relative fitness.

DOI: https://doi.org/10.7554/eLife.48548.007

copies generated during meiosis are rapidly removed by purifying selection (*Figure 1—figure supplement 3*). In short, none of the simulations generated the empirically observed pattern of lower TE loads in asexual than sexual strains. In a second step, we therefore allowed TE activity rates to evolve over time, by introducing a modifier allele that increases excision rates. The allele has no direct fitness effect, so it can only be fixed in a population via genetic hitchhiking. In simulations that included the modifier allele, the modifier spreads rapidly to fixation in asexual strains, because it is associated with genomes that have fewer TE copies, and therefore have a higher relative fitness. As a consequence, TE activity rates decrease in asexual populations (*Figure 1—figure supplement 4*). By contrast, the modifier cannot spread as rapidly in sexual populations because recombination constantly breaks up the association between the modifier and less TE loaded backgrounds. By allowing for the evolution of TE activity rates in our simulations, we were able to identify parameter values representative for yeast that result in simulations with a very close fit to our empirical results

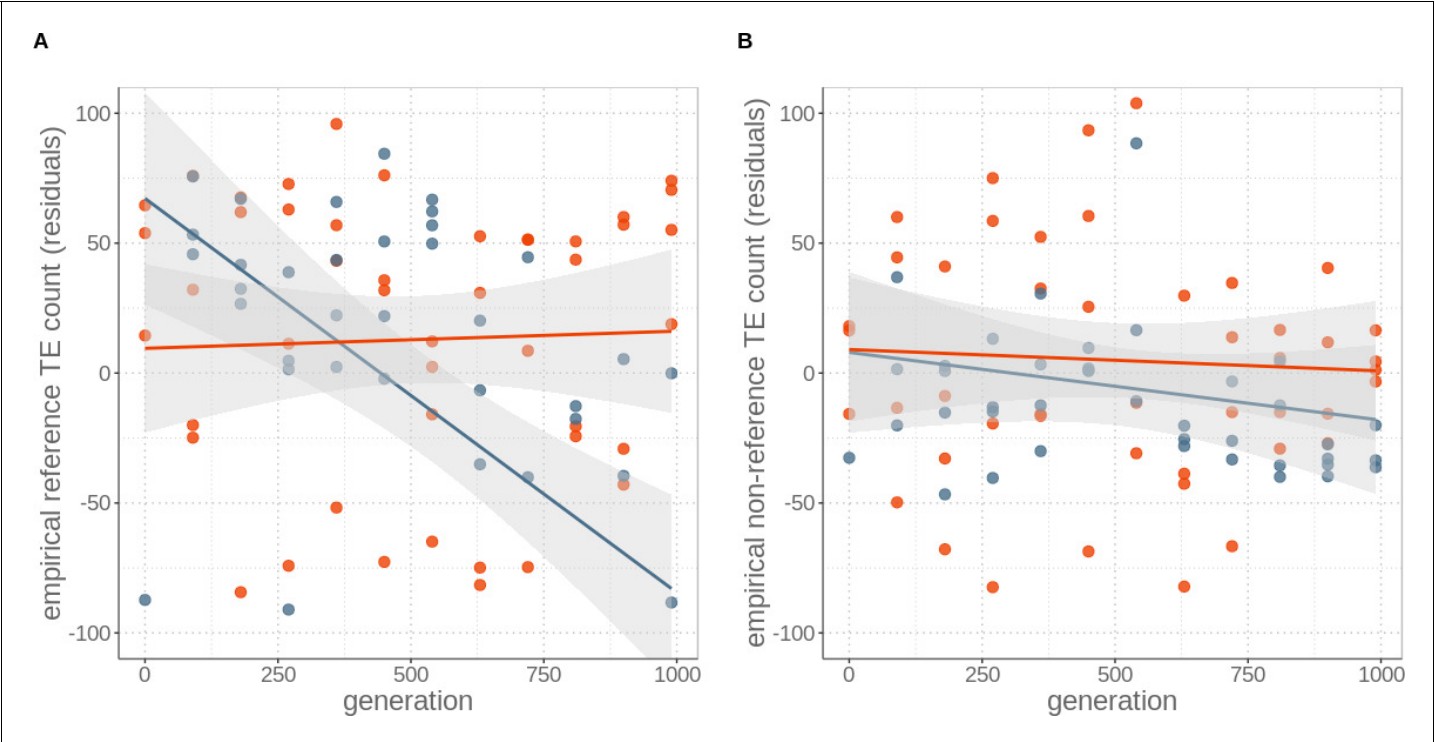

**Figure 2.** Decrease of insertions in asexuals over time is largely due to loss of 'ancestral' reference insertions (A) rather than novel insertions (B). Count of all TE insertions, irrespective whether full-length TE, solo LTR, truncated elements or other types in genomes of four replicates of sexual (red) and asexual (blue) yeast strains over 1000 generations of experimental evolution. Numbers are expressed as residuals, since TE detection probability depends on sequencing coverage. Lines represent linear regression for sexuals (red) and asexuals (blue) and the grey areas represent 95% CI.
DOI: https://doi.org/10.7554/eLife.48548.008

(*Figure 1B*, *Supplementary file 2B*). These analyses thus corroborate our empirical findings that a likely mechanism driving genomic TE load reduction in asexual yeast strains is the rapid evolution of increased TE excision rates. A similar effect would be expected if our modifier acted on transposition rather than excision rates, since the net TE activity depends on the relative rates of transposition vs excision. However, our empirical results do not suggest major differences in transposition rates between sexual and asexual yeast strains. In combination with our findings that, in the absence of TE activity evolution, sexual strains always lose TEs faster than asexual ones, the empirical results are best explained by an increase in TE excision rates under asexuality (*Figures 1* and *2*).

Our study shows that sexual reproduction permits the maintenance of TEs in *S. cerevisiae*, while in its absence, TE loads decrease, likely via the evolution of TE activity rates. The findings are consistent with empirical findings of low TE activity in old asexuals (*Bast et al., 2016*) and the idea that TEs should evolve to be benign in asexual species, because the evolutionary interests of TEs and their host genome are aligned (*Charlesworth and Langley, 1986*). While the exact mechanisms causing TE activity change in the asexual yeast populations cannot be assessed in the empirical data, our simulations suggest that there is some form of TE defense mechanism (a 'modifier locus') that either segregates in the ancestral yeast strain used in the experiments or repeatedly appeared de novo during experimental evolution. Independently of the exact mechanism, we confirm that TE loads do not increase, but decrease, in asexual populations. This contrasts with the hypothesis that most asexual species are evolutionarily short lived because they are driven to extinction via negative consequences of accumulating TE copies (*Arkhipova and Meselson, 2005*). Instead, sex is at the root of the evolutionary success of parasitic TEs.

## Materials and methods

### Yeast experimental evolution

We used data generated in a previous study based on experimental evolution of the yeast *S. cerevisiae* (for in-depth details see *McDonald et al., 2016*). In short, 12 different strains were initiated from the same pool of ancestral strains (derived from haploid W303 strains) and kept under constant conditions. Sexual reproduction in yeast depends on the presence of two separate mating types. Only individuals with different mating types can fuse and go through meiosis. Asexual reproduction occurs through budding. For the experiment, six haploid strains consisting of mating type a (MATa) and six haploid strains of mating type α (MATα), were grown over 990 generations. Of these, four strains were grown exclusively asexually (two of MATa, two of MATα), while the eight others (four of MATa, four of MATα) were mixed for mating events every 90 generations, resulting in four sexual strains. Paired-end Illumina reads were generated for each of the 12 different strains every 90 generations during 990 generations (for a total of 11 sequencing events per strain). Read numbers per sample ranged from 12,775 to 10,270,312, averaging 2,964,869 reads per sample, with a total of 818,303,966 reads. Details of the read data can be found at BioProject PRJNA308843 and in the original study (*McDonald et al., 2016*).

### Data processing

The genome of the haploid W303 *S. cerevisiae* strain was retrieved from *Lang et al. (2013)*. All Illumina paired-end raw reads of the 12 replicate strains generated in *McDonald et al. (2016)* were downloaded from the SRA (BioProject identifier PRJNA308843). Raw reads were quality filtered by first removing adapter sequences (with the script used in the original study; *McDonald et al., 2016*; provided by Daniel P Rice, Harvard University), followed by removing the first 10 bases and quality trimming using trimmomatic v0.33 (*Bolger et al., 2014*) with parameters set to LEADING:3 TRAILING:3 HEADCROP:10 SLIDINGWINDOW:4:15 MINLEN:36. Additionally, non-overlapping paired-reads were constructed in silico from the subset of the original paired-reads that were overlapping, as a prerequisite to run the insertion detection pipeline. For this, overlapping reads (on average overlapping by 16 bp) were merged using PEAR v0.9.6 with standard parameters (*Zhang et al., 2014*). Merged reads were split in half and 20 bp deleted from each read at the overlapping ends using the fastx_toolkit v 0.0.13.2 (*Hannon Laboratory, 2010*). This resulted in mean read lengths of 72 bp. These 'artificial' non-overlapping read pairs were afterwards merged with the read set fraction that was non-overlapping.

### Overall transposable element load

A *S. cerevisiae* specific, curated and updated TE library that contained all consensus sequences of all TE families found in this species is available from *Carr et al. (2012)*. With this library, we identified TE content and specific copy insertions in the W303 genome using RepeatMasker v4.02 (*Smit et al., 2015*) with parameters set to -nolow -gccalc -s -cutoff 200 -no_is -nolow -norna -gff -u -engine rmblast. For overall TE load estimates, the fraction of reads mapped to TEs out of total mappable reads was calculated. For this, the TE library was appended to the masked W303 genome and all reads for all strains and generations were mapped using BWA v0.7.13 with standard parameters (*Li, 2013*). For all strains, mean per-base coverage was checked with bedtools genomecov v2.26 (*Quinlan and Hall, 2010*), upon which the asexual strain sample 3D-90 was excluded from all further analyses, as coverage was lower than one-fold for this sample. Following this analysis, stat-reads from the PopoolationTE2 v1.10.04 program (*Kofler et al., 2016*) was utilised to extract the number of total mapped reads and reads mapped to TEs. For statistics, a permutation ANOVA with the formula lm(coverage ~generation*mode) was utilised; for details see github repository (*Bast and Jaron, 2019*, copy archived at https://github.com/elifesciences-publications/reproductive_mode_TE_dynamics).

### Specific transposable element insertions

To detect specific reference (present in the reference genome) and non-reference TE insertions in all samples, the McClintock pipeline was utilised (*Nelson et al., 2017*). This pipeline combines six different, benchmarked programs in a standardised fashion. McClintock was run with the non-

overlapping read set, the curated TE library, and the W303 assembly using default parameters. The nonredundant insertions output file per sub-program was collected. Next, we utilised a custom python script to collect all information on insertions detected by all different programs and counted insertions with evidence from different programs only once.

To identify full-length TEs and solo LTR insertions from the McClintock custom filtered output, we tagged insertions by length according to the typical *TY* TE properties found in *S. cerevisiae* (i.e. a full TE is a combination of internal sequence and two LTRs within a 500 bp range; solo LTRs are between 220 and 420 bp; see *Supplementary file 1*). Because TE insertion detection was influenced by the coverage, coverage was taken into account when calculating the number of insertions, by adding it as random factor (coverage effect p<0.001, generation effect p=0.006, reproductive mode effect p=0.033, and interaction between generation and mode p<0.001; permutation ANOVA with the formula lm(counts ~ coverage+generation*mode); for details see github repository, *Bast and Jaron, 2019*). We then calculated the number of lost TEs in asexual strains from the regression slope in asexuals after correcting for coverage (i.e. computing residuals) over 1000 generations, with 50 full-length TEs in the ancestor. To additionally check for a bias due to coverage differences between sexual and asexual strains, we randomly subsampled read data for each sample corresponding to the mean coverage of the asexual strains for each generation (*Figure 1—figure supplement 2*).

## Modelling

To model TE dynamics in yeast we adjusted an individual based, forward in time simulator by *Dolgin and Charlesworth (2006)*. We extended the model to include sexual cycles via fusion of two haploid individuals and recombination, with on average one crossover on each of the 16 modelled chromosomes (yeast has 16 chromosomes; *Goffeau et al., 1996*; *McDonald et al., 2016*). Each chromosome carries 200 loci that are potential targets for a TE insertion. A simulation is initiated with a single individual with 50 TEs randomly placed in the 3200 loci of the genome. The founder individual then populates clonally the whole simulated deme of explicitly simulated 100,000 individuals. With currently available computational resources, there was no need to scale deterministic parameters of the model as was done in the original study by *Dolgin and Charlesworth (2006)*. To account for mutations during this phase, we ran 20 burn-in generations of transposition and excision cycles on every individual separately without applying selection. One generation in the simulation consists of a round of selection and reproduction with transposition occurring during reproduction, followed by excision. The relative fitness $w_n$ of an individual carrying $n$ TEs was modelled as $w_n = exp\left(-an - \frac{1}{2}bn^2\right)$, where $a$ and $b$ are parameters representing the strength of selection and the strength of epistatic interactions between TEs respectively (*Dolgin and Charlesworth, 2006*). The simulation was then continued for 990 generations. We performed 10 replicates of each simulation. Using the average TE load in the population measured every 10 generations, we fitted a linear model to estimate average TE loss across the ten replicates of each simulation. Parameters were derived from yeast experimental measurements and simulations were run with perturbation in the surrounding parametric space (see *Supplementary file 2A*). We further explored the effects of different transposition rates during meiosis vs asexual reproduction, but this did not change the dynamics even for meiotic transposition rates that were not biologically plausible (up to 10% of TEs transposing during meiosis). The last extension included the introduction of an unlinked, general modifier allele increasing the excision rates of all elements by the same amount. The parameters related to this extension are the initial frequency of the modifier allele and the excision rate increases when the modifier allele is present (see *Supplementary file 2B*). See the code documentation for details.

## Data availability

Raw read data of the experiment are available at SRA (BioProject identifier PRJNA308843).

## Code availability

The code used for both the analyses of empirical data and for the theoretical prediction of TE dynamics together with explanations are available online at https://github.com/KamilSJaron/reproductive_mode_TE_dynamics (copy archived at https://github.com/elifesciences-publications/reproductive_mode_TE_dynamics).

## Acknowledgements

We thank Michael J McDonald, Daniel P Rice and Michael M Desai for providing the experimental evolution raw data and for helpful explanations. We further thank Patrick Tran Van for setting up the insertion pipeline, Daniel L Jeffries for providing the TE wrapper script, Beatriz Navarro Dominguez for improving the empirical analyses R script and Deborah Charlesworth, Brian Charlesworth, Graham Coop and Laurent Keller for discussions and comments on the manuscript. This study was supported by DFG research fellowships (grant numbers BA 5800/1–1 and BA 5800/2–1 to JB) and by funding from the University of Lausanne and Swiss SNF (grant numbers PP00P3_170627 and PP00P3_139013 to TS).

## Additional information

### Funding

| Funder | Grant reference number | Author |
| --- | --- | --- |
| Deutsche Forschungsgemeinschaft | BA 5800/1-1 | Jens Bast |
| Schweizerischer Nationalfonds zur Förderung der Wissenschaftlichen Forschung | PP00P3_17062 | Tanja Schwander |
| Schweizerischer Nationalfonds zur Förderung der Wissenschaftlichen Forschung | PP00P3_139013 | Tanja Schwander |
| Deutsche Forschungsgemeinschaft | BA 5800/2-1 | Jens Bast |

The funders had no role in study design, data collection and interpretation, or the decision to submit the work for publication.

### Author contributions

Jens Bast, Conceptualization, Data curation, Formal analysis, Funding acquisition, Investigation, Visualization, Writing—original draft, Project administration, Writing—review and editing; Kamil S Jaron, Data curation, Software, Formal analysis, Investigation, Visualization, Methodology, Writing—original draft, Writing—review and editing; Donovan Schuseil, Formal analysis, Investigation, Visualization, Writing—review and editing; Denis Roze, Software, Investigation, Writing—review and editing; Tanja Schwander, Conceptualization, Supervision, Funding acquisition, Writing—original draft, Project administration, Writing—review and editing

### Author ORCIDs

Jens Bast (iD) https://orcid.org/0000-0003-0017-3860
Kamil S Jaron (iD) https://orcid.org/0000-0003-1470-5450

### Decision letter and Author response

Decision letter https://doi.org/10.7554/eLife.48548.015
Author response https://doi.org/10.7554/eLife.48548.016

## Additional files

### Supplementary files

• Supplementary file 1. *S. cerevisiae* TY elements and the sizes (in bp) of internal regions and LTRs and the size boundaries used for filtering.
DOI: https://doi.org/10.7554/eLife.48548.009

• Supplementary file 2. (**A**) Explored parameter space of the simulations as pertinent for yeast (empirically determined values in bold). Selection_a and selection_b are selection coefficients for linear fitness effects and epistasis, respectively. Lost_TEs refers to the total number of TE lost after

1000 generations (averaged over ten replicates). (**B**) Explored parameter space for simulations including a modifier allele. Highlighted is the simulation closest to empirical observations. Init_f is the frequency of the modifier at the start of the simulations. Selection_a and selection_b are selection coefficients for linear fitness effects and epistasis, respectively. Lost_TEs refers to the total number of TE lost after 1000 generations (averaged over ten replicates). The bold lines refer to parameter combinations that generate results close to the observed empirical values.

DOI: https://doi.org/10.7554/eLife.48548.010

• Transparent reporting form

DOI: https://doi.org/10.7554/eLife.48548.011

## Data availability

Raw read data of the experiment are available at SRA (BioProject identifier PRJNA308843). All data processing and analyses scripts as well as the simulator together with explanations are available at https://github.com/KamilSJaron/reproductive_mode_TE_dynamics (copy archived at https://github.com/elifesciences-publications/reproductive_mode_TE_dynamics).

The following previously published dataset was used:

| Author(s) | Year | Dataset title | Dataset URL | Database and Identifier |
|---|---|---|---|---|
| McDonald MJ, Rice DP, Desai MM | 2016 | Saccharomyces cerevisiae (baker's yeast) | https://www.ncbi.nlm.nih.gov/bioproject/PRJNA308843/ | NCBI BioProject, PRJNA308843 |

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
