## [Decision Letter]

Thank you for submitting your article "Asexual reproduction drives the reduction of transposable element load" for consideration by *eLife*. Your article has been reviewed by three peer reviewers, including Graham Coop as the Reviewing Editor and Reviewer #1, and the evaluation has been overseen by Diethard Tautz as the Senior Editor. The following individual involved in review of your submission has agreed to reveal their identity: Brian Charlesworth (Reviewer #2).

The reviewers have discussed the reviews with one another and the Reviewing Editor has drafted this decision to help you prepare a revised submission.

The reviewers and I appreciated the analysis and results. I have included the two reviewers' comments below.

Please respond point by point to the reviewer comments below. The major comments that definitely need addressing are:

1) Moderate the causal language concerning selection and increase in excision rate, the evidence for this is indirect at best. See reviewer 1.

2) Reviewer 2 raises a concern about how the non-independence of temporal samples from the same population replicate are dealt with. One suggestion from the reviewer/editor discussion was to just use the difference between the initial and end points. The authors more generally need to be clearer about the analysis performed, e.g. give the formulae for the linear models run.

3) Reviewer 2 raises some questions about full-length and non-reference TEs that need to be addressed.

4) The McDonald experiment is of pooled sequencing, thus you are averaging over population-frequencies of TE at each locus. While TE-calling from the short-read data, and asexuality, it might be very hard to do anything with the frequency of TEs at each locus I thought it would be helpful to more fully acknowledge that the current analysis (I think) confounds the number TE loci in the genome and the frequency of the TE at each locus.

Reviewer #2:

This paper presents an interesting analysis of the population dynamics of transposable elements (TEs) in long-term experimental populations of budding yeast, comparing sexual and asexual populations. The data are analysed in the light of simulations, which extend an earlier study by Dolgin and Charlesworth, 2007, to match more closely the properties of the yeast populations. The overall conclusion is that excisions of the LTR retrotransposons (presumably by recombination between the LTRs; this could have been investigated) cause a decline in TE copy number in the asexuals, whereas the sexuals seem to stay more or less in equilibrium.

This is one of the few studies of this type of problem, and illustrates the difficulty of making generalisations about the fates of TEs in populations with different mating systems, as the authors make clear. I have to take their bioinformatic analyses on trust, as I lack the relevant expertise, but they seem to have done a thorough job of these. Overall, this is a nice paper.

My only criticism is that they emphasis the possible role of selection for an increase in excision rate in explaining their results, but I could not see that they have presented any solid evidence that this has actually happened. In an asexual population, all kinds of hitchhiking will be going on, so any increase in frequency of a modifier of excision rate could simply be due to such an effect, although of course one would not expect consistency across replicate populations. They need to make it clearer what the evidence for such selection actually is; it's not obvious to me that there has been an increase in excision rates.

Reviewer #3:

Bast et al. address how sexual reproduction affects transposable element (TE) accumulation in paired sexual and asexual lineages of yeast. As noted in the manuscript, much of the literature surrounding the issue of the impact of sex on TE accumulation is confounded by deeper evolutionary timescales, different effective population sizes, and changes in mating system. Utilizing existing data of yeast experimental evolution lines to get around these issues, Bast et al. find that TEs are spread through sex, and driven towards extinction in asex lines. The manuscript is clearly written and easy to follow. Specific comments below.

My understanding from a brief glance at the McDonald et al., 2016 paper these data come from suggests that the data represent eight separate lineages, such that each point should be connected through the time series in Figures 1 and 2. I'm not familiar enough with the statistics involved, but if each is an independent lineage through time, does this need to be included in analyses? It also seems that some of the uncertainty in TE genotyping could be addressed using replicate lineages – if a full-length copy is 'excised,' we don't expect to see it again in later generations of that replicate.

- Main text, fifth paragraph: Throughout the manuscript, the use of the term 'excision' might confuse readers more familiar with different types of TEs. For DNA transposons (cut and paste), this refers to a complete or near complete removal via transposition. In this manuscript, 'excision' is used for solo LTR formation via unequal recombination. To avoid confusion, perhaps 'solo LTR formation' could be used as an alternative to excision.

- Main text, fifth paragraph: It would be useful to add that solo LTR formation (excision) removes the protein coding genes that allow transposition, to make it clear why we care about full-length copies.

- Main text, sixth paragraph: I am a bit confused on the numbers of full length TEs identified. The second to last sentence states only 24 of 50 full length copies can be detected, but the last sentence states asex decreases to 41 full length copies. Although identifying TEs is really difficult, I am concerned that there could be deletions or excisions in these 26 non-assayable copies that overwhelm the signal observed. Could a total count of transposition-competent copies be tracked through time by something akin to the first coverage based approach (like Figure 1—figure supplement 1), using internal protein coding regions? Or is there a way to explain what's happening with these non-assayable copies? Are they always the same copies in every individual?

Doesn't a constant number of non-reference copies through time (Figure 2B) mean that there is an increased transposition rate in asex through time? To me it feels like since both the asex full length copies (presumably the active copies) and reference copies are being removed through time, this means the per-element transposition rate is going up. Again, this could be addressed by identifying how many non-reference TEs are the same non-reference TE copies (at the same loci) through time (although maybe the low coverage of some samples precludes this?). But it's hard for me to think in residuals, so maybe this isn't impacting things, and the slightly more negative trend in asex in 2B reflects the effect of a higher excision rate on generating fewer new copies.

- Subsection “Modelling”: I need a little more information on the TE annotation. How are full length copies being defined from the RepeatMasker output? Those which contain any internal protein coding sequence? A length cutoff relative to the reference db?

---

## [Author Response]

Please respond point by point to the reviewer comments below. The major comments that definitely need addressing are:1) Moderate the causal language concerning selection and increase in excision rate, the evidence for this is indirect at best. See reviewer 1.

We changed the language accordingly throughout the manuscript.

2) Reviewer 2 raises a concern about how the non-independence of temporal samples from the same population replicate are dealt with. One suggestion from the reviewer/editor discussion was to just use the difference between the initial and end points. The authors more generally need to be clearer about the analysis performed, e.g. give the formulae for the linear models run.

The strain replication is taken into account in our model, as we included generation in the permutation ANOVA. We clarified this in the text and the formulae are now given in the manuscript. Additionally, we calculated the statistics for start and end points. For details, please see concern one of reviewer three below.

3) Reviewer 2 raises some questions about full-length and non-reference TEs that need to be addressed.

Detailed replies to these two questions are given at the reviewer’s comments below.

4) The McDonald experiment is of pooled sequencing, thus you are averaging over population-frequencies of TE at each locus. While TE-calling from the short-read data, and asexuality, it might be very hard to do anything with the frequency of TEs at each locus I thought it would be helpful to more fully acknowledge that the current analysis (I think) confounds the number TE loci in the genome and the frequency of the TE at each locus.

We added clarifications to the manuscript. See also the detailed reply to this comment at the section of reviewer three.

Thank you again for your constructive comments.

Reviewer #2:[…] My only criticism is that they emphasis the possible role of selection for an increase in excision rate in explaining their results, but I could not see that they have presented any solid evidence that this has actually happened. In an asexual population, all kinds of hitchhiking will be going on, so any increase in frequency of a modifier of excision rate could simply be due to such an effect, although of course one would not expect consistency across replicate populations. They need to make it clearer what the evidence for such selection actually is; it's not obvious to me that there has been an increase in excision rates.

We rephrased the wording accordingly throughout the manuscript to avoid suggesting a strong implication of selection.

Reviewer #3:[…] My understanding from a brief glance at the McDonald et al., 2016 paper these data come from suggests that the data represent eight separate lineages, such that each point should be connected through the time series in Figures 1 and 2. I'm not familiar enough with the statistics involved, but if each is an independent lineage through time, does this need to be included in analyses? It also seems that some of the uncertainty in TE genotyping could be addressed using replicate lineages – if a full-length copy is 'excised,' we don't expect to see it again in later generations of that replicate.

The strain replication is taken into account in our model, as we included generation in the permutation ANOVA, and there is only one data point per strain per time. We focused on the generation effect, because we are interested in the temporal dynamics of TE loads. We did not separate the strains in the graphics to correspond with the underlying statistics.

We clarified this now by adding the ANOVA formula to the Materials and methods, which was previously only shown in the github repository. See the last paragraph of the subsection “Specific transposable element insertions”.

Additionally, to address the issue, we calculated statistics of a paired-test of the start (generation 90) and end-points (generation 990) per lineage/strain: Sexuals *P* = 0.84; Asexuals *P* = 0.03.

- Main text, fifth paragraph: Throughout the manuscript, the use of the term 'excision' might confuse readers more familiar with different types of TEs. For DNA transposons (cut and paste), this refers to a complete or near complete removal via transposition. In this manuscript, 'excision' is used for solo LTR formation via unequal recombination. To avoid confusion, perhaps 'solo LTR formation' could be used as an alternative to excision.

Indeed, this might be confusing. We added clarifications to the Introduction (main text, fifth paragraph).

- Main text, fifth paragraph: It would be useful to add that solo LTR formation (excision) removes the protein coding genes that allow transposition, to make it clear why we care about full-length copies.

We added the clarifications (main text, fifth paragraph). Thank you for helping to make the manuscript more clear.

- Main text, sixth paragraph: I am a bit confused on the numbers of full length TEs identified. The second to last sentence states only 24 of 50 full length copies can be detected, but the last sentence states asex decreases to 41 full length copies. Although identifying TEs is really difficult, I am concerned that there could be deletions or excisions in these 26 non-assayable copies that overwhelm the signal observed. Could a total count of transposition-competent copies be tracked through time by something akin to the first coverage based approach (like Figure 1—figure supplement 1), using internal protein coding regions? Or is there a way to explain what's happening with these non-assayable copies? Are they always the same copies in every individual?

We know that the ancestral reference strain has 50 full length copies. With the insertion detection method (McClintock), we can only identify 24 in the starting strains (generation 0, which should be about 50). These are not necessarily the same in the following generations. This is because with illumina data it is very hard to identify all insertions and not possible to track them through time. This is why we take the slope of the linear regression to estimate the loss from the original 50 copies. Like this, we can compare it to the simulations.

Additionally, our first approach estimates total TE content, using the fraction of mappable reads that map to the TE copies (including both internal coding and flanking LTR regions), is independent of the need to detect specific insertions and gives very similar results (slopes; Figure 1—figure supplement 1). This approach does not allow for discriminating specific insertions or truncated TE copies (which can still include internal coding regions). However, this approach is representative of mostly losing internal regions, as both internal and LTR were included in the TE library, such that loss of internal regions should make a bigger difference (solo LTRs still covered by reads).

Doesn't a constant number of non-reference copies through time (Figure 2B) mean that there is an increased transposition rate in asex through time? To me it feels like since both the asex full length copies (presumably the active copies) and reference copies are being removed through time, this means the per-element transposition rate is going up. Again, this could be addressed by identifying how many non-reference TEs are the same non-reference TE copies (at the same loci) through time (although maybe the low coverage of some samples precludes this?). But it's hard for me to think in residuals, so maybe this isn't impacting things, and the slightly more negative trend in asex in 2b reflects the effect of a higher excision rate on generating fewer new copies.

These are very interesting questions, but unfortunately, we cannot disentangle whether there is faster TE turnover in each sexual genomic background, or the same TE insertions are maintained through time with the data available in the study. This is because pooled sequencing lowers the probability of TE detection in specific individuals, such that we cannot analyse turnover or activity of specific elements in specific genomic backgrounds. What we analysed is the presence of TE insertions at a given generation, meaning they are present in enough genotypes in the population to be picked up in relatively low-coverage illumina sequencing. We clarified this in the Results (main text, sixth paragraph).

- Subsection “Modelling”: I need a little more information on the TE annotation. How are full length copies being defined from the RepeatMasker output? Those which contain any internal protein coding sequence? A length cutoff relative to the reference db?

We identified full-length LTR elements based on the TE insertion length information from the mcClintock output, that was filtered by a custom script to collect all information from the various McClintock output files. These include both internal coding sequences and flanking LTR regions. See subsection “Specific transposable element insertions”, last paragraph.